# Plant Poly(ADP-Ribose) Polymerase 1 Is a Potential Mediator of Cross-Talk between the Cajal Body Protein Coilin and Salicylic Acid-Mediated Antiviral Defence

**DOI:** 10.3390/v15061282

**Published:** 2023-05-30

**Authors:** Nadezhda Spechenkova, Viktoriya O. Samarskaya, Natalya O. Kalinina, Sergey K. Zavriev, S. MacFarlane, Andrew J. Love, Michael Taliansky

**Affiliations:** 1Shemyakin-Ovchinnikov Institute of Bioorganic Chemistry of the Russian Academy of Sciences, 117997 Moscow, Russia; rysalka47@gmail.com (N.S.); kalinina@belozersky.msu.ru (N.O.K.);; 2Belozersky Institute of Physico-Chemical Biology, Lomonosov Moscow State University, 119991 Moscow, Russia; 3The James Hutton Institute, Invergowrie, Dundee DD2 5DA, UK

**Keywords:** poly ADP-ribosylation (PARylation), nucleolus, Cajal bodies, tobacco rattle virus, PARP inhibitor, salicylic acid

## Abstract

The nucleolus and Cajal bodies (CBs) are sub-nuclear domains with well-known roles in RNA metabolism and RNA-protein assembly. However, they also participate in other important aspects of cell functioning. This study uncovers a previously unrecognised mechanism by which these bodies and their components regulate host defences against pathogen attack. We show that the CB protein coilin interacts with poly(ADP-ribose) polymerase 1 (PARP1), redistributes it to the nucleolus and modifies its function, and that these events are accompanied by substantial increases in endogenous concentrations of salicylic acid (SA), activation of SA-responsive gene expression and callose deposition leading to the restriction of tobacco rattle virus (TRV) systemic infection. Consistent with this, we also find that treatment with SA subverts the negative effect of the pharmacological PARP inhibitor 3-aminobenzamide (3AB) on plant recovery from TRV infection. Our results suggest that PARP1 could act as a key molecular actuator in the regulatory network which integrates coilin activities as a stress sensor for virus infection and SA-mediated antivirus defence.

## 1. Introduction

Poly ADP-ribosylation (PARylation) is a well-known highly conserved posttranslational protein modification. During PARylation, ADP-ribose moieties are typically covalently (or non-covalently) added sequentially to target proteins, leading to decoration of the proteins with poly(ADP-ribose) (PAR) chains of varying length and branching morphology [1,2]. PARylation is carried out by poly(ADP-ribose) polymerases (PARPs), a process which uses nicotinamide adenine dinucleotides (NAD+) as a donor of ADP-ribose for PAR synthesis [1,2]. The resultant PARylated proteins, including PARP itself and other protein targets, act as regulators of diverse key biological processes such as DNA repair, programmed cell death and responses to biotic and abiotic stresses and also participate in various other cellular signalling pathways. Once the PARylated proteins have fulfilled their regulatory functions, poly(ADP-ribose) glycohydrolases (PARGs) hydrolyse the PAR-chains attached to the proteins, which liberates free PAR or ADP-ribose. Subsequently, free PAR is cleaved into AMP and ribose-5-phosphate via the activity of nucleoside diphosphate linked to some moiety-X (NUDIX) hydrolases [1,2].

The PARP protein family consists of 17 mammalian proteins, with PARP1 as the best studied and most abundant member which accounts for approximately 90% of the PARP activity [3]. This is a nuclear enzyme whose activities, along with associated PAR-mediated events, are all intimately tied to subnuclear compartments such as Cajal bodies (CBs) and the nucleolus. The nucleolus has well-described roles in rDNA transcription, rRNA processing and ribosome assembly [4,5,6,7], and it is physically and functionally associated with CBs. CBs participate in the maturation of certain nuclear RNAs, and the assembly, modification and transport of RNP particles of different classes (e.g., spliceosomal small nuclear RNPs, small nucleolar RNPs, telomerase and U7 snRNP) (for a review, see [8,9,10,11,12]). In addition, the nucleolus and CBs are also involved in crucial cell functions such as cell growth and death, the cell cycle and stress responses [12,13].

In *Drosophila* cells, unmodified PARP1 molecules are normally localised in the nucleolus and bind to chromatin. However, PARP1 automodified by PARylation as well as other PARylated proteins have been shown to interact non-covalently via PAR polymers with coilin, the major scaffolding protein of CBs [11,14]. It is thought that PARP1 facilitates the relocalisation of these PARylated proteins from the nucleolus and chromatin to CBs for subsequent reactivation/recycling via the activity of PARG, which cleaves off PAR groups. As CBs play an important role in diverse stress responses, it seems likely that the interaction of PARP1/PARylated proteins with coilin/CBs may act as a mechanism which regulates PARylation levels in order to activate various responses to pathogen attacks, abiotic stresses and developmental cues [15,16].

In contrast to mammals, the Arabidopsis genome contains only three genes encoding PARP proteins. Two of them, PARP1 and PARP2, are nuclear and have been demonstrated to be bona fide poly(ADP-ribose) polymerases which catalyse PARylation reactions [1,17,18]. Based on the phylogenetic relationships, structural similarities and domain architectures, these proteins closely resemble their respective PARP1 and PARP2 animal counterparts. Plant PARP3 has no counterpart in animals, and its functional role remains largely unknown. With regards to plant PARP1 and PARP2, although assumptions concerning their functions have often been inferred by analogy with animal PARPs, the plant-specific roles of these proteins have become apparent [1,2]. In particular, plant PARP1 and PARP2 proteins have been linked to plant responses to biotic and abiotic stresses [1,17,19].

PARylation events are known to be a significant factor involved in the development of responses to a wide range of DNA- and RNA-containing human and animal viruses [20,21]. Interestingly, PARylation may differentially modulate virus infections either by enhancing [22] or inhibiting [20] virus propagation. There are several lines of evidence pointing towards the suggestion that PARylation may also play a role in plant virus interactions [23,24,25]. For example, we observed a significant upregulation of various PAR-degrading enzymes such as PARG and NUDIX in response to infection of potato plants with potato virus Y (PVY). According to these changes, PAR accumulation was significantly increased upon PVY infection compared to non-infected plants. It is thus conceivable to suggest that the PVY–plant interactions are at least partially regulated by PARylation activities [24,25]. However, the mechanisms underpinning functional links between PARylation and plant responses to viral infections remain unknown. New insights into the molecular and cellular functions of PARylation in plant responses to virus attack may come from studies showing that CBs and their major scaffolding protein, coilin, affect a variety of interactions between viruses and host plants [15,16,26,27,28,29,30]. For example, transgenic coilin gene RNA silenced plants which had knocked down coilin gene expression and were devoid of CBs, and had differing responses to different viruses relative to wildtype plants. Coilin/CBs may be recruited by some viruses for enhancement of their replication, but interestingly these cell components may in some cases, such as with tobacco rattle virus (TRV; [28]), act to suppress infection.

TRV is a bipartite, positive-strand RNA virus. TRV RNA1 encodes the replication and movement of proteins including the 16K cysteine-rich silencing suppressor [31]. RNA2 encodes the coat protein as well as one or more additional proteins that are involved in the transmission by nematodes and are not required for virus infectivity in plants [31]. TRV infection has several stages. The virus accumulates to high levels in the initially infected leaves. However, after one to two weeks these levels may decline, and the infected plants enter a recovery phase, so that in newly emerging leaves, the disease symptoms become much milder and virus replication is reduced to very low levels [28,32,33]. Previously, plant recovery from TRV was attributed to RNA silencing-mediated plant defence [33].

However, we have recently found that, in contrast to WT plants, transgenic coilin knockdown (KD) plants did not recover from TRV infection and exhibited high levels of virus accumulation, although antiviral RNA silencing in these plants was as active as in WT plants [28]. This suggests that in addition to RNA silencing, coilin may also promote another host defence mechanism operating in virus-infected leaves. Indeed, we have shown that an essential step in this mechanism is the stimulation of nucleolar relocalisation of coilin induced by the TRV 16K protein. This viral protein is known to localise to the cytoplasm, nucleus and also the nucleolus [34,35]. Futhermore, 16K-mediated nucleolar redistribution of coilin is accompanied by activation of salicylic acid (SA)-dependent defence responses, which acts to restrict TRV systemic infection and induces plant recovery from TRV [29]. SA is an important plant hormone that acts as a major signalling factor in triggering systemic acquired resistance (SAR) [36]. However, the molecular mechanisms underpinning functional links between 16K-induced nucleolar localisation of coilin and SA-mediated defence responses that suppress TRV systemic infection remain unknown.

Here, we investigate the role of the plant PARP1 protein as a potential mediator of cross-talk between TRV16K-induced nucleolar localisation of coilin and SA-dependent defence against TRV. We show that coilin physically interacts with PARP1 and redistributes it to the nucleolus, and that this interaction is modulated by the TRV 16K protein. The 16K/coilin-mediated increase in the nucleolar retention of automodified PARP1 prevents it trafficking from the nucleolus to CBs for PAR cleavage and recycling. In addition, the pharmacological and genetic inhibition of PARP activity reveals a positive correlation between PAR over-accumulation and the SA-induced defence responses causing plant recovery from TRV. These findings uncover a hitherto unrecognised interplay between fundamental biological mechanisms underpinning coilin/CB dynamics and PARylation in regulating plant antiviral immunity.

## 2. Materials and Methods

### 2.1. Transgenic Plants and TRV-Based Constructs

Coilin-silenced (KD) *Nicotiana benthamiana* plants with approximately 80% reduction in the coilin transcript level were generated in a previous study [28] via transgenic expression of a hairpin RNA construct, which was designed to avoid “off-target” silencing [28]. The construct expressing the TRV 16K protein used in this work was generated earlier (for details, see [29]). The TRV and TRV∆16K infectious cDNA clones were generated and described previously [37,38]. Plasmids were transformed into *Agrobacterium tumefaciens* strains LBA4404 and C58C1 for delivery into *N. benthamiana* plants by agroinfiltration [28,29]. For TRV infections, separate *Agrobacterium* cultures containing constructs for RNA1 and RNA2 were prepared and mixed in a 1:1 ratio as described previously [29]. The cultures expressing TRV RNAs or the TRV 16K protein alone were infiltrated to the underside of a leaf using a 2 mL syringe without a needle.

### 2.2. RNA Extraction and Northern Blot Analysis

Leaf tissues (1 to 2 g) were frozen in liquid nitrogen and ground to a fine powder in a mortar and pestle, and RNA was extracted with TRI REAGENT (Sigma Aldrich; St. Louis, MO, USA) according to the manufacturer’s recommendation. RNA was suspended in 50 μL DEPC-treated water and resolved using electrophoresis in 1% agarose gel and then electro-blotted to Hybond N membrane prior to UV cross-linkage using a StrataLinker (Stratagene; La Jolla, CA, USA). For analysis of TRV accumulation, we used as a probe a BstEII fragment (positions 5345 to 6792) of pTR7116 corresponding to part of TRV RNA1 [33]. The DNA fragment was labelled by random-primed incorporation of ^32^P-dCTP using a Invitrogen^TM^ random-priming DNA labelling system (Invitrogen; Waltham, MA, USA). As a loading control, equal fractions of each sample were resolved on a 1% agarose gel and stained with ethidium bromide (EtBr). Similar results were obtained in four independent experiments.

### 2.3. Real Time Quantitative RT-PCR (RT-qPCR)

Total RNA was isolated as described above. Residual DNA was removed by treating the RNA with RNase-free DNase I (Invitrogen). Aliquots of DNase-treated RNA were transcribed into cDNA using the SuperScriptTM First-Strand Synthesis System for RT-PCR (Invitrogen), in conjunction with either an oligo-dT primer (for host mRNAs) or a TRV RNA1 specific primer (see Appendix A). The primer pairs for SYBR green based real-time PCR analysis of TRV RNA1, PARP1 and PR1-a (designed using PRIMER EXPRESS software; Applied Biosystems, Foster City, CA, USA) are listed in Appendix A. Primer concentrations giving the lowest threshold cycle (C_t_) value were used in further analysis along with 10-fold dilutions in sterile water of the first-strand cDNA reaction mixes. These components were mixed with reagents in the QuantiTect^TM^ SYBR^®^ Green PCR kit (Qiagen; Crawley, UK) according to the manufacturer’s instructions, prior to loading into an ABI PRISM 7700 Sequence Detection System (Applied Biosystems) for product amplification using the following reaction temperatures and durations: 95 °C for 15 min, followed by 40 cycles of 94 °C for 15 s, 60 °C for 30 s and 72 °C for 30 s. The Ct value for PVY and each mRNA was normalized to reference gene mRNAs encoding Ubiquitin3 (UBI3) and the ribosomal protein L23 (L23); (Appendix A).

### 2.4. Immunolabelling and Confocal Imaging Analysis

Leaf tissues of *N. benthamiana* plants were fixed in 3.7% (*v*/*v*) formaldehyde and 5% (*v*/*v*) dimethyl sulfoxide in PHEM buffer (60 mM PIPES, 25 mM HEPES, 5 mM EGTA, 2 mM MgCl_2_, pH 6.9) for 2 h. After fixation, the leaf tissues were digested with cell wall-degrading enzymes (1% cellulase, 0.1% pectolyase and 0.1% bovine serum albumin in PHEM buffer) for 2 h, incubated in 1% (*v*/*v*) Triton X-100 for 20 min and treated with ice-cold methanol for 10 min. The samples were washed three times with PHEM after each step. PARP1 was localized with immunofluorescence techniques [26,27], using primary rabbit antibodies to a KLH-conjugated synthetic peptide derived from *Arabidopsis thaliana* PARP1 (1:100; Agrisera; Vännäs, Sweden), respectively. The primary antibodies were visualized by Alexa Fluor 488-conjugated anti-rabbit secondary antibodies (1:500; Invitrogen). Subcellular localisation of the proteins was determined using a TCS SP2 confocal laser scanning microscope (Leica Microsystems; Deerfield, IL, USA). The acquired images were processed using Photoshop CC (Adobe Inc, San Jose, CA, USA). To locate nuclei, the leaf tissues were infiltrated with PBS containing 4,6-diamidino-2-phenylindole (DAPI). Quantification of the levels of accumulation of PARP1 in the nucleolus versus nucleoplasm was carried out using ImageJ software [39]. For visualization of the nucleolus and CBs, we used fibrillarin fused to mRFP as a nucleolar and CB marker which was delivered into cells via *Agrobacterium* mediated expression [27] (see Appendix A).

### 2.5. Co-Immunoprecipitation and Western Blotting

Nuclear extracts were prepared from 0.5 g of plant leaves essentially as described elsewhere [40]. Leaves were ground to a fine powder in liquid nitrogen and mixed with 2 volumes of lysis buffer (20 mM Tris-HCl, pH 7.4, 25% glycerol, 20 mM KCl, 2 mM EDTA, 2.5 mM MgCl_2_, 250 mM sucrose and 1 mM PMSF). The homogenate was filtered twice through Miracloth to remove plant debris before centrifugation at 1500× *g* for 10 min. The solution was sonicated, and the resulting nuclear extract was used directly for immunoprecipitation [14]. Extracts were incubated with Protein-G Sepharose 4B (Sigma Aldrich) on a rotating platform, and beads were collected by spinning for 1 min, 2000× *g*. Appropriate amounts of antibodies were added to the lysates and incubated for 4 h at 4 °C. The antibody to PARP (1:100) was the same as described in previous section, and coilin (1:1000) were the same as described above. The rabbit antibodies to TRV 16K and coilin were produced as described previously [29,35]. Then, Protein-G Sepharose 4B was added to the lysates and incubated overnight at 4 °C with rotation. Beads were washed in the lysis buffer. Bound proteins were eluted by Laemmli loading buffer and analysed by western blotting. The total proteins were resolved by SDS-PAGE and subsequently transferred to a nitrocellulose membrane (Protran Whatman, Sigma–Aldrich, Zwijndrecht, The Netherlands), and then probed with the primary anti-coilin or anti-PARP antibodies. The secondary antibody used was goat anti-rabbit serum. The reactions were visualised using the Amersham ECL Western Blotting System (GE Healthcare Life Sciences; Uppsala, Sweden).

### 2.6. Immunological Detection of Poly ADP-Ribose (PAR)

For the isolation of plant nuclei and nuclear protein extraction, the CelLytic PN kit (Sigma Aldrich) was used [41]. The protein (5 µg) was analysed for PAR accumulation levels by ELISA using purified monoclonal antibody (Trevigen) to PAR as the capture reagent, a rabbit anti-PAR antibody (Trevigen; Gaithersburg, MD, USA) as the detecting agent, and a goat anti-rabbit antibody conjugated with alkaline phosphatase (Sigma Aldrich) as the reporter [42]. PAR polymer (Trevigen) was used as a positive control.

### 2.7. 3-Aminobenzamide (3AB) Treatment

Plant leaves were first sprayed with either 0.6% dimethyl sulfoxide (DMSO; no PARP inhibitor) or 2.5 mM 3AB (Sigma Aldrich) (dissolved in 0.6% DMSO) 2 h before inoculation with or without TRV, and then the treatment was repeated every 48 h post inoculation for three weeks (until the end of the experiment).

### 2.8. Expression and Purification of the Recombinant Coilin and Far-Western Blot Analysis

The construct expressing the plant 6xHis-coilin gene was described previously [43]. The recombinant coilin protein was purified under denaturing conditions using Ni-NTA Qiagen agarose (Qiagen), according to the manufacturer’s protocol. For analysis of the PARP1–coilin interaction, recombinant human PARP1 protein (Sigma Aldrich) and BSA (as a negative control) were electrophoresed through a polyacrylamide gel, transferred to nitrocellulose membrane, incubated with or without recombinant coilin produced as described above and probed with anti-coilin rabbit polyclonal antiserum (1:6000) and goat anti-rabbit serum (1:15,000). The reactions were visualised using the BCIP/NBT system.

### 2.9. Virus-Induced Silencing of PARP1 Expression

Two fragments of NbPARP1 were amplified using primers 2838 (TGGATGGGATAGCCTCTCAG) and 2839 (GAGTGCTCCAAAAAGCATCC) to produce a 440 nt fragment 1 at nucleotides 491 to 968, and primers 2832 (GGACTAAGAATTGCTCCTCCA) and 2703 (CCGCTTATAATTAAACCTCAC) to produce a 410 nt fragment 2 at the 3′ terminus of the NbPARP1 gene. These were inserted in antisense orientation separately into the PVX vector (pGR106) genome [44] to produce two different NbPARP1 silencing constructs. Empty PVX vector (PVX-C) was used as negative control. Three lower leaves of four young (4–5 leaf stage) *N. benthamiana* plants were infiltrated with *A. tumefaciens* GV3101 cultures (OD_595_ = 0.1) carrying a PVX control or PVX-PARP1 VIGS constructs. Plants untreated with *Agrobacterium* (-PVX) were used as an additional control. At five days after agroinfiltration, plants were inoculated with TRV. Ten days later, leaf samples were collected from either the inoculated leaves or apical tip (systemically infected) leaves. The four leaf samples from plants with each treatment were combined before RNA was extracted and analysed as described above. Two separate PVX-NbPARP1 VIGS constructs made in this work exhibited similar effects on *PARP1* gene expression.

### 2.10. Measurement of Endogenous SA

Free SA and SA-b-glucoside (conjugated SA) were extracted essentially as described by [45]. Leaf tissues were ground in liquid nitrogen, mixed with 70% aqueous EtOH (*v*/*v*) and o-anisic acid (OAA; internal control) and centrifuged. The supernatant was transferred to a new centrifuge tube. The remaining pellet was resuspended in 90% MeOH and re-centrifuged as above. Then the two supernatants were combined, and the alcohol (EtOH and MeOH) was evaporated. The remaining aqueous solution was added to a mixture of ethyl acetate and cyclohexane and centrifuged for phase separation. The organic phase contained free SA. The aqueous phase containing the methanol-soluble conjugated SA was diluted with equal volumes of 8M HCl and heated for 1 h at 80 °C for hydrolysis of conjugated SA. SA and SA hydrolysate quantification was carried out using high-performance liquid chromatography with fluorimetric detection using SA and OAA calibration curves [45]. Additional details can be found in [29].

### 2.11. Foliar Application of SA

For SA treatment experiments, plants were sprayed with either a control solution of 0.11% (*v*/*v*) ethanol or 1 mM SA dissolved in 0.11% (*v*/*v*) ethanol at 1 day prior to virus/mock inoculation and then daily for three consecutive days. All data are presented as mean values from four biological replicates (from each of four experiments); each replicate was composed of samples from three plants collected together (two leaves per plant).

### 2.12. Callose Staining

To detect callose, leaves were cleared with 95% ethanol overnight, stained with 150 mM K2P04 (pH 9.5), 0.01% aniline blue for 2 h and examined with a Leica MZ FL III Fluorescence microscope [25].

## 3. Results

### 3.1. PARP1 Gene Expression Is Unaffected by TRV Infection

In previous work, we showed that TRV initially accumulated to high levels in the inoculated and already emerged uninoculated (old systemically infected) leaves of wild type (WT) *N. benthamiana* plants. However, 8–10 d post-inoculation (dpi), these plants exhibited a clear recovery with a marked reduction in virus accumulation and symptom production in the newly emerging leaves. In contrast, coilin-silenced (KD) transgenic plants infected with TRV or WT plants infected with a 16K deletion mutant of TRV (TRV∆16K) did not recover from infection and exhibited persistent severe systemic symptoms and high rates of virus accumulation in all (old and newly emerging systemically infected) leaves [28,29]. This indicates that both the TRV 16K protein and coilin are involved in antiviral resistance mechanisms to (i.e., recovery from) TRV infection.

To examine a possible link between this coilin-mediated antiviral defence in *N. benthamiana* and PARP1 activities, we first analysed the effect of TRV infection on *PARP1* gene expression by RT-qPCR assay and found that PARP1 mRNA levels were not significantly affected by TRV in *N. benthamina* plants (Figure 1a,b). Moreover, no noteworthy differences were observed in PARP1 mRNA levels in TRV-infected WT (recovered) or coilin KD (unrecovered) plants, or in WT plants infected with either TRV (recovered) or TRV∆16K (unrecovered). Thus, no correlation between TRV accumulation rates and *PARP1* gene expression was detected (Figure 1a,b).

### 3.2. Interaction between Coilin and PARP1

Next, we tested if plant coilin, like animal coilin, can interact with PARP1. Using an antibody against coilin for co-immunoprecipitation, we found that coilin interacted with PARP1 in all samples, each of which contained substantial amounts of both of these proteins (WT plants infected or uninfected with TRV or TRV∆16K). In contrast, in coilin-deficient KD plants, precipitation of PARP1 by the anti-coilin antibody was not observed (or produced only a very weak signal), although it was clearly detected in the leaf total protein extract (input) (Figure 2a). In the reciprocal assay, the anti-PARP1 antibody was able to efficiently co-precipitate coilin from extracts of all plants containing coilin (WT plants infected or uninfected with TRV or TRV∆16K but not from samples deficient in coilin (TRV-infected coilin KD plants), suggesting that these two proteins specifically interact with each other. Interestingly, the anti-16K antibody was also able to efficiently co-precipitate PARP1 with 16K but only in the presence of substantial amounts of coilin (WT plants); in coilin-deficient KD plants, co-precipitation was absent or poor (Figure 2a). In contrast, interaction of PARP1 with coilin does not require 16K, since it could occur not only in TRV-infected plants but also in mock-inoculated WT plants and WT plants infected with TRV∆16K.

Taken together these data confirm that plant coilin, like animal coilin, can form a complex with PARP1, which may also include the 16K if all these three proteins are present in infected tissues. Furthermore, direct interaction between PARP1 and coilin was demonstrated in vitro (by far-western blot assay; Figure 2b).

### 3.3. Effect of TRV Infection on the Intracellular Localisation of PARP1 and Accumulation of PARylated Proteins

Given that 16K can interact with coilin and mediate its relocalisation to the nucleolus [29], it could be expected that in this situation PARP1 as an interacting partner of coilin may also be redistributed to the nucleolus. To confirm this hypothesis, we examined the localisation of PARP1 in plants using a fluorescently labelled anti-PARP1 antibody. Similar to what is known for *Drosophila* PARP1, in healthy (mock inoculated) WT plants, PARP1 was distributed between the nucleolus, nucleoplasm (presumably binding to chromatin) and CBs (Figure 3a,b; Appendix A). In contrast, in unrecovered leaves of WT plants infected with TRV, the distribution of PARP1 was found to be markedly altered, with an over-accumulation in the nucleolus and much less in the nucleoplasm and CBs (Figure 3a,b; Appendix A). These effects required the presence of the 16K protein, because they were not observed in plants infected with TRV∆16K or in recovered leaves of TRV-infected plants (TRV-R) containing no or very little TRV (Figure 3a,b; Appendix A). Moreover, transient expression of the 16K protein alone also led to the nucleolar redistribution of PARP1, suggesting that the 16K protein is the TRV product which is necessary and sufficient for this process. Increased accumulation of PARP1 in the nucleolus also required substantial amounts of coilin, because its nucleolar localisation was not enhanced by TRV infection or the 16K protein in coilin-deficient (KD) plants (Figure 3a,b; Appendix A).

If PAR removal and recycling of PARP1 and other PARylated proteins in plants occur in CBs (as was suggested for *Drosophila* cells [14]), it could be expected that, following TRV-induced nucleolar sequestration of PARP1, levels of PAR (associated with unrecycled PARP1 target proteins) might be significantly increased. Indeed, it was shown that PAR accumulated to higher levels in *N. benthamiana* plants infected with TRV (in infected unrecovered leaves containing 16K) compared to plants infected with the TRV∆16K mutant virus (Figure 3c), which did not to retain PARP1 in the nucleolus (Figure 3a,b). Accordingly, coilin KD plants infected with TRV in which PARP1 was not trapped in the nucleolus (Figure 3a,b) did not display an increase in PAR levels (Figure 3c).

It thus appears that the TRV16K-mediated nucleolar relocalisation of coilin in TRV-infected cells is a prerequisite for nucleolar retention of PARP1 and consequent over-accumulation of PARylated proteins. It was therefore decided to investigate whether such coilin-mediated changes in PARP1 intracellular localisation, functions and activities could lead to activation of a plant systemic defence system(s) resulting in plant recovery from TRV.

### 3.4. Effect of Deficiency in PARP Activity on TRV Infection

To determine a potential causal relationship between the over-accumulation of PARylated proteins induced by the nucleolar retention of PARP1 in response to TRV infection and plant recovery from the virus, we analysed the effect of the pharmacological PARP inhibitor 3-aminobenzamide (3AB), which targets the conserved enzymatic active site [47,48]. We found that treatment of TRV-infected WT *N. benthamiana* plants with 3AB substantially increased systemic symptom severity and levels of virus accumulation, and prevented recovery from the virus (for at least three weeks from the initial infection) (Figure 4a,b). There was no obvious effect of 3AB on the physical appearance of uninfected (mock-inoculated) plants (Figure 4a), indicating that 3AB directly affects TRV-plant interactions. In addition, high-level accumulation of PARylated proteins induced by TRV in WT plants (Figure 4c) was decreased by treatment with 3AB to levels observed in plants showing no recovery (i.e., TRV-infected coilin KD plants or plants infected with TRV∆16K; compare Figure 4c to Figure 3c).

Therefore, to confirm our findings obtained with the 3AB PARP inhibitor, we used an additional approach based on virus-induced gene silencing (VIGS) of PARP1 expression. VIGS constructs were made by insertion of fragments of the PARP1 gene into a potato virus X (PVX) expression vector (pGR106; [44,49]), giving PVX-NbPARP1. This was used in conjunction with an empty PVX vector (PVX-C) negative control in subsequent experiments. It was found that the expression of PARP1 mRNA and accumulation of PARylated proteins were significantly suppressed by PVX-NbPARP1 infection (starting at 7 dpi) compared with control plants infected with PVX-C (Figure 5a,b). With regard to the effect on TRV infection, VIGS of PARP1 led to enhanced TRV accumulation and TRV-induced symptom severity (stunting, leaf chlorotic mottle, leaf curling) in newly emerging leaves and as such prevented recovery from the virus when compared with the control (Figure 5c,d), confirming the pharmacological data that PARP1 participates in defence responses against TRV.

Collectively, these data support the idea that it is a change in PARP1 shuttling activity leading to over-accumulation of PAR (associated with PARP1 target proteins) in TRV-infected leaves that may subsequently activate host defence and result in plant recovery.

### 3.5. The Role of PARP1 in Plant SA-Mediated Defence Responses against TRV

Previously we found that coilin-mediated plant defence responses to TRV infection operate through the SA signalling pathway [29], a well-known key factor of plant antiviral defence mechanisms [50,51]. Given that coilin and PARP1 interact with each other in *N. benthamiana* plants, we examined whether and how PARP1 may regulate SA-dependent pathways.

First, our results showed that the substantial increase in endogenous concentrations of both free SA and conjugated SA (SA-b-glucoside) [52,53] during TRV infection in inoculated and both old and newly emerging systemically infected leaves (compared to mock-inoculated plants) [29] coincided with nucleolar retention of PARP1, over-accumulation of PARylated proteins and TRV recovery (Figure 1b and Figure 3). Moreover, the PARP inhibitor 3AB, which prevented plant recovery from TRV infection (Figure 3), also significantly suppressed TRV-induced over-accumulation of free and conjugated SA in all inoculated and systemically infected leaves (Figure 6a,b). At the same time, the relatively low levels of free and conjugated SA in mock-inoculated plants were unaffected by 3AB treatment (Figure 6a,b).

The SA-mediated defence responses are usually associated with induction of a number of pathogenesis-related (PR) proteins [54]. Although PR proteins do not display any direct antiviral activity, the activation of PR protein expression (and PR-1a, in particular) is considered as an important hallmark of SAR [51,55]. Callose deposition is another SA-associated hallmark of plant defence responses, which regulates permeability of plasmodesmata, cytoplasmic channels between plant cells and hence cell connectivity during a defence response [56,57]. Transcription rates of the *PR-1a* gene as well as deposition of callose were substantially enhanced following TRV infection in both inoculated and non-inoculated (old and new systemically infected) leaves (Figure 7a,b). In contrast, as an indication that PARP1 is involved in specific SA-mediated defence against TRV infection, we found that deficiency in PARP activity caused by 3AB treatment in TRV-infected plants (Figure 4c) correlated well with reduced levels of *PR-1a* gene expression and callose deposition (Figure 7a,b).

Finally, we showed that exogeneous application of SA subverted the negative effects of the PARP inhibitor 3AB on the SA signalling pathway and strongly enhanced expression of the *PR-1a* gene and also increased callose deposition (Figure 7a,b). Our data suggest that 3AB lowers SA levels, and this reduction likely impinges on downstream SA signalling. 3AB treatment does not block components of the SA signalling pathway downstream of SA since exogenous application of SA can re-establish SA pathway activity in the presence of 3AB.

Consistently, symptom severity and accumulation of TRV RNA in newly emerging systemically infected leaves of these plants were dramatically decreased by SA treatment, and as a result these plants recovered from infection by the virus (Figure 7c,d).

Taken together, these data suggest that the mechanism of plant recovery from TRV involves a functional crosstalk between PARP-mediated PARylation and SA-mediated defence signalling pathways.

## 4. Discussion

Previously we have shown that in TRV-infected *N. benthamiana* plants, coilin recognises, binds to and uses the nucleolus-localised virus-encoded 16K protein for its relocalisation from CBs to nucleoli [29]. In WT plants, these events are accompanied by enhanced accumulation of SA and activation of SA-mediated signalling pathways, and trigger restriction of TRV systemic infections which is manifested as plant recovery [29]. However, the molecular mechanisms underlying the functional link between nucleolar coilin-16K interactions and the enhanced accumulation of SA, which concomitantly stimulates SA-mediated defence pathways, remained unknown.

In this paper, we demonstrate that PARP1 is a second interacting partner of plant coilin. The 16K-induced nucleolar relocalisation of coilin occurring upon TRV infection results in an increase in the nucleolar retention of PARP1, thereby preventing it trafficking from the nucleolus to CBs for PAR cleavage and recycling. The pharmacological (3AB inhibitor) and genetic (VIGS) studies have revealed a positive correlation between over-accumulation of PARylated proteins and the defence responses which facilitate plant recovery from TRV (low TRV accumulation in upper newly emerging leaves). We therefore hypothesize that coilin may play a surveillance role, recognising certain virus proteins (such as TRV 16K), mediating changes in PARP1 localisation pattern and overaccumulation of PARylated proteins, and then promoting defence. It is known that in plants, as in animals, poly(ADP-ribosyl)ation plays an important role in responses to biotic and abiotic stresses [1,2,47]. Our experiments using the PARP inhibitor 3AB suggest that PARP1 operates in plant defence, responding to TRV infection by activating SA accumulation and triggering the SA-signalling pathway. Thus, both coilin KD and PARP deficiency in plants prevent activation of SA-dependent defence pathways, which results in increased disease symptoms and virus spread to newly emerging leaves of the infected plant. Although it could not be completely excluded that coilin and PARP1 operate independently by different mechanisms, it seems more likely that both these proteins act in concert to initiate the antiviral defence mechanism. For example, these proteins can interact with each other, and both coilin KD and deficiency in PARP activity caused by 3AB exhibit similar effects on SA accumulation and expression of SA-responsive genes (Figure 6 and Figure 7). Furthermore, the deficiency of SA caused by the PARP inhibitor 3AB or PARP1 VIGS did not affect PAR over-accumulation during TRV infection (Figure 4) but prevented plant recovery. This suggests that the PAR over-accumulation apparently caused by the retention of PARP1 in the nucleolus precedes and possibly triggers elevations in SA accumulation and subsequent activation of the SA-mediated defence pathway.

We propose a model demonstrating the integral connection between the coilin-mediated nucleolar retention of PARP1 and its biological function in plant host defence against TRV (Figure 8). In healthy plants, PARP1 modifies the function and subcellular localisation of a variety of nuclear “target” proteins (acceptors) by attaching chains of ADP ribose (PAR) to them. To “re-set” these target proteins, PARP1 shuttles them from both the nucleolus and chromatin to CBs, presumably for PAR removal and recycling. Coilin in these plants is located within CBs and the nucleoplasm and, being an interacting partner of PARP1, plays an important role in its trafficking to CBs. In TRV-infected plants, the TRV 16K protein enters the nucleus and interacts with coilin (in CBs and/or the nucleoplasm) redistributing it to the nucleolus. In the nucleolus, coilin binds to auto-modified PARP1 preventing it and other PAR-modified proteins from being trafficked to CBs for re-cycling, resulting in over-accumulation of PAR (in the form of PARylated proteins). Our pharmacological (3AB inhibitor) and genetic (VIGS) studies have revealed a positive correlation between PAR over-accumulation and the enhanced production of SA and expression of SA-responsive genes (exemplified by *PR-1a*), which also coincides with plant recovery from TRV (low TRV accumulation in upper newly emerging leaves). Thus, PARP1 can act as a mediator to link coilin activities and SA-mediated antivirus defence.

A logical corollary of this would be that coilin is a sensor for TRV infection which can modify PARP1 activity in order to activate the SA-dependent defence signalling pathway. A key element of this antiviral defence mechanism is the redistribution of coilin to the nucleolus, where it normally does not locate, except in the case of TRV infection where it is redistributed by the TRV 16K protein. In animal cells, coilin interacts with the survival of motor neuron (SMN) protein complex and targets it to CBs for maturation of spliceosomal small nuclear RNPs, and the WRAP53 protein is essential for this process [58,59]. Interestingly *SMN*- or *WRAP53*-knockdown results in the loss of CBs and nucleolar redistribution of coilin [58,59]. Some other conditions such as treatment with leptomycin B [60], an inhibitor of nuclear export, can also lead to nucleolar redistribution of coilin. Future studies are required to examine if depletion of plant homologues of SMN [61,62] and WRAP53 [62] or applications of leptomycin B may redistribute coilin to nucleoli, which may cause nucleolar retention of PARP1 and over-accumulation of PARylated proteins, increased SA concentrations and attenuation of TRV or other plant virus infections.

A versatile antiviral defence response in plants is well accepted to be based on RNA silencing [33,63,64]. In *N. benthamiana* plants infected by TRV, RNA silencing mechanisms also operate [28,29,33], but they may not be solely responsible for recovery of the plant from TRV infection [28,29]. We propose that, in order to restrict TRV systemic infection, the plant deploys a combination of both RNA silencing mechanisms and the coilin-PARP1-associated SA-dependent defence responses, which can function in parallel to give effective protection against TRV infection.

These data may also have implications for a variety of other plant virus infections that exhibit a recovery phenomenon. Indeed, similar to TRV, some other plant viruses such as tomato black ring virus and tomato spotted wilt virus that enter a recovery phase do not only activate RNA silencing but also induce other defence responses, for example, callose deposition [65]. Collectively, these observations support the idea that plants can employ multiple defence mechanisms to restrict viral replication and movement, including hormone-mediated defence, gene silencing and immune receptor signalling [66]. A better understanding of these mechanisms will certainly assist in the development of novel approaches to control viral pathogens for sustainable agriculture.

There is a high degree of cross-kingdom functional conservation of PARP and sub-nuclear bodies. Both PARP and CBs have been implicated in responses to a wide range of biotic and abiotic stresses [12,13] and are known to be important factors involved in the development of defence responses to some human, animal and plant viruses [20,21,22,26,27,29,30]. It will be of substantial interest to explore the extent to which CBs/coilin and PARP1 operate as novel interactive players in defence-signalling (sensing and triggering) pathways against these stresses in plants and other eukaryotic systems.

## Figures and Tables

**Figure 1 viruses-15-01282-f001:**
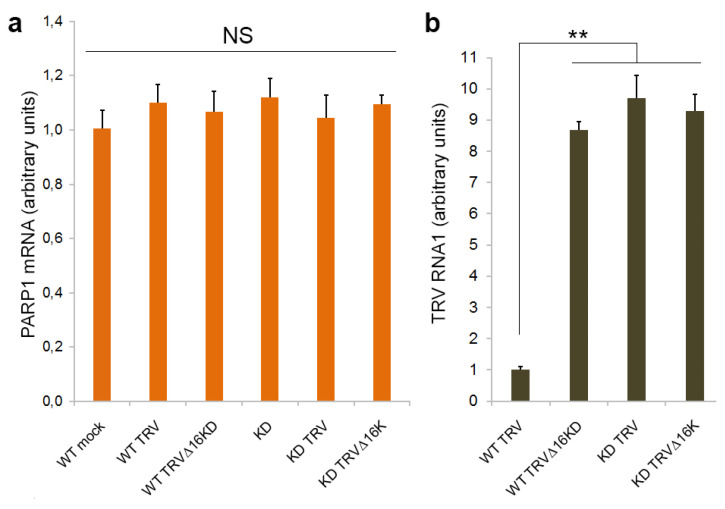
Effect of TRV infection on *N. benthamiana PARP1* (*NbPARP1*) gene expression at 14 dpi. Accumulation of PARP1mRNA (**a**) and TRV RNA1 used as an indicator of TRV multiplication [33] (**b**) (measured by RT-qPCR) in newly emerged leaves of WT and coilin KD *N. benthamiana* plants infected with or without (mock-inoculated) TRV or TRV∆16K. PARP1 mRNA and TRV RNA1 expression levels were normalized to those of internal *N. benthamiana* controls, *UBIQUITIN3* gene (*UBI3)* and *60S ribosomal protein 23* gene (*L23*) [46]. Statistical analysis was performed on four independent biological replicates. Data are mean ± SD. Each replicate was composed of samples from three plants pooled together (two leaves per plant). Analysis of variance and Tukey’s HSD post hoc tests were performed on the RT-qPCR data. NS, not significant; ** *p* < 0.01.

**Figure 2 viruses-15-01282-f002:**
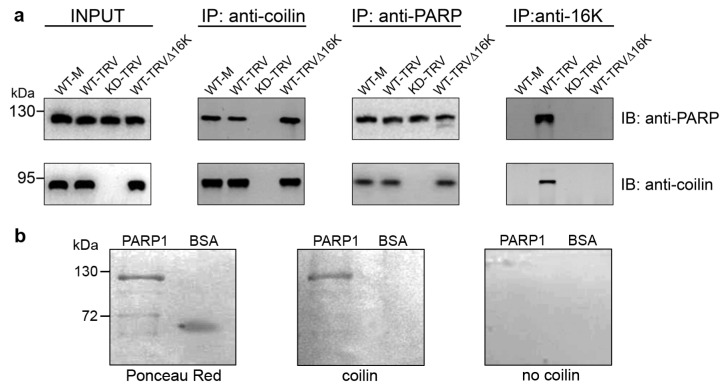
Interaction of PARP1 with coilin. (**a**) Co-immunoprecipitation of PARP1, coilin and TRV 16K protein. Protein extracts were prepared at 7 dpi from older systemically infected leaves (third and fourth leaves above the inoculated leaf) of WT or coilin KD plants infected with or without (mock-inoculated, M) TRV or TRV∆16K, as indicated. Proteins in the lysate prior to immunoprecipitation (IP) are shown on the left (input). Anti-coilin antibodies (anti-coilin) were used to co-immunoprecipitate coilin and PARP. Anti-PARP antibodies were used to co-immunoprecipitate PARP and coilin. Antibodies to 16 K (anti-16K) were used to co-precipitate 16K, coilin and PARP. Proteins were detected by western blot analysis (immunoblotting, IB) using anti-PARP and anti-coilin antibodies. Positions of molecular mass markers are on the left. Images have been cropped for presentation. Uncropped images are presented in Appendix A. (**b**) Far-western blot analysis of the in vitro interaction between coilin and commercially-sourced PARP1. Bovine serum albumin (BSA) was used as a negative control. The left blot was stained with Ponceau red; the middle and right blots were incubated with and without the recombinant coilin, respectively (as indicated), and then probed with anti-coilin antiserum. Positions of the molecular mass markers are indicated on the left.

**Figure 3 viruses-15-01282-f003:**
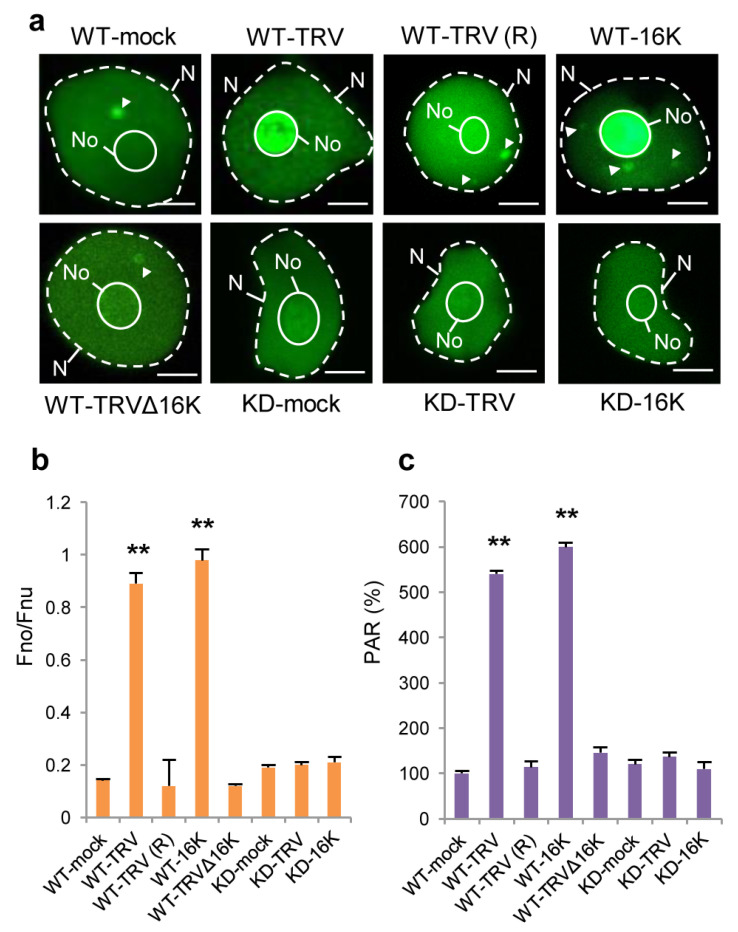
The interaction between coilin and PARP1 induced by TRV results in partial nucleolar sequestration of PARP1 and over-accumulation of PARylated proteins. (**a**) Representative images of intranuclear distribution of PARP1 (immunofluorescent staining using primary rabbit anti-PARP1 antibody and secondary fluorescent anti-rabbit antibody, green ) in WT and coilin KD *N. benthamiana* plants infected with or without (mock-inoculated) TRV or TRV∆16K were taken at 7 dpi in older systemically infected leaves (third and fourth leaves above the inoculated leaf) or at 14 dpi in recovered newly emerging leaves of WT plants systemically infected with TRV [seventh and eighth leaves above the inoculated leaf; WT-TRV(R)] or at 3 days post-agroinfiltration (dpa) in leaves agroinfiltrated with a construct expressing the 16K protein (WT-16K or KD-16K). N, nuclei; No, nucleoli; CBs are shown by arrows. Scale bars, 5 µm. (**b**) Quantification of results presented in (**a**). The ratio of nucleolar fluorescence to nucleoplasmic fluorescence (Fno/Fnu) was averaged for at least 100 cells in three independent experiments. Data are mean ± SD. Analysis of variance and Tukey’s HSD post hoc tests were performed on the data obtained. ** *p* < 0.01. (**c**) Accumulation of PARylated proteins measured by ELISA using rabbit anti-PAR polyclonal antibody, in plants described in (**a**). Data are mean ± SD, n = 6 from three independent experiments. Analysis of variance and Tukey’s HSD post hoc tests were performed on the data obtained. ** *p* < 0.01.

**Figure 4 viruses-15-01282-f004:**
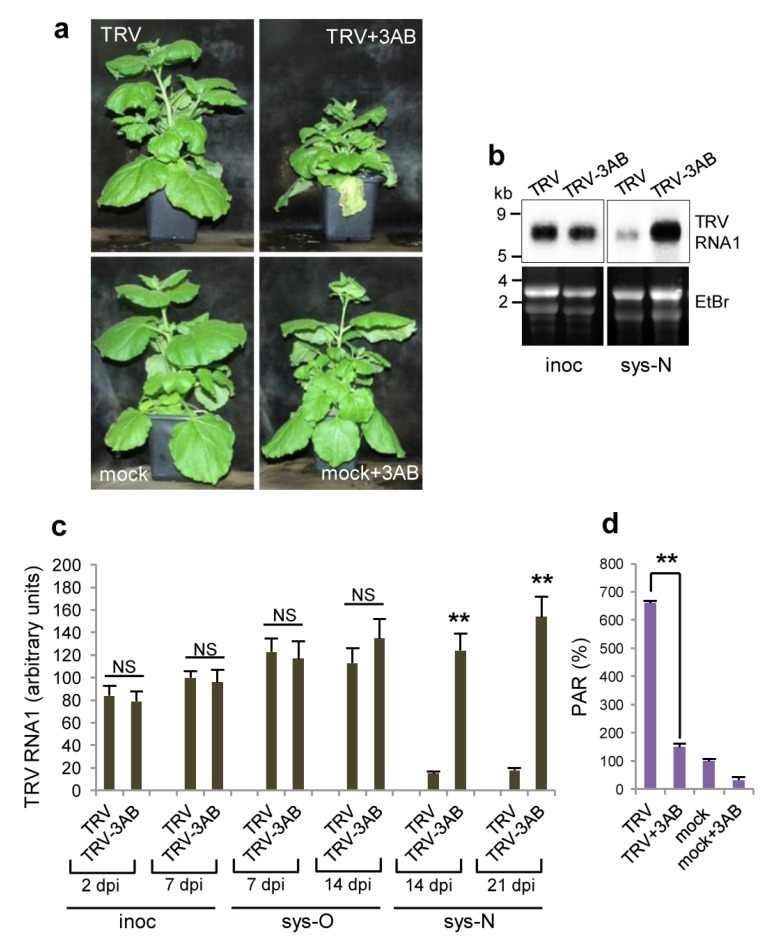
Effect of 3-aminobenzamide (3AB) on the development of TRV infection and accumulation of PARylated proteins in WT *N. benthamiana* plants. (**a**) Symptoms induced in TRV-infected and mock-inoculated plants treated with or without 3AB. (**b**) Northern blot analysis of TRV RNA1 in inoculated (inoc, 7 dpi) and newly emerging systemically infected leaves (seventh and eighth leaves above the inoculated leaf; sys-N, 14 dpi). The positions of RNA size markers are indicated on the left. Ethidium bromide (EtBr)-stained rRNA (bottom panel) is shown as a loading control. Images have been cropped for presentation. Uncropped images are presented in Appendix A. (**c**) Accumulation of TRV RNA1 (measured using RT-qPCR) in inoculated (inoc; 2 and 7 dpi), older systemically infected leaves (third and fourth leaves above the inoculated leaf; sys-O, 7 and 14 dpi) and newly emerging systemically infected leaves (seventh and eighth leaves above the inoculated leaf; sys-N, 14 and 21 dpi). TRV RNA1 expression levels were normalized to those of the internal controls, *UBIQUITIN3* gene *(UBI3)* and *60S ribosomal protein 23* gene *(L23)*. (**d**) Accumulation of PARylated proteins was measured by ELISA using rabbit anti-PAR polyclonal antibody, in TRV- systemically infected or mock-inoculated plants treated with or without 3AB (older third and fourth leaves above the inoculated leaf). Data are mean ± SD, *n* = 6 from three independent experiments. Analysis of variance and Tukey’s HSD post hoc tests were performed on the data obtained. ** *p* < 0.01; NS, non-significant (**c**,**d**). These data could suggest that a change in PARP1 shuttling activity leading to over-accumulation of PAR (associated with PARP target proteins) in TRV-infected leaves subsequently activates host defence and results in plant recovery. However, given that pharmacological PARP inhibitors including 3AB may not only affect the activity of canonical PARPs, but also have off-target effects [1,2,48], results of pharmacological experiments to infer PARP function in plants should be verified by the genetic inhibition of PARP activity.

**Figure 5 viruses-15-01282-f005:**
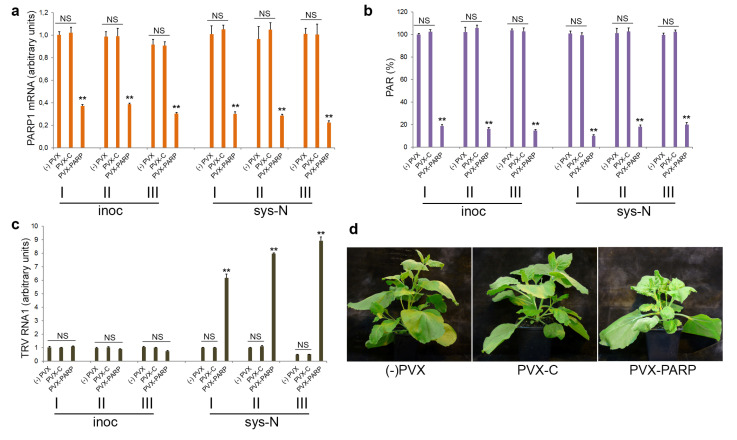
Effect of virus-induced silencing of *PARP1* (*NbPARP1*) expression on TRV infection. Two separate PVX-PARP1 VIGS constructs made in this work (see Materials and Methods) exhibited similar effects on *NbPARP1* gene expression (**a**), accumulation of PARylated proteins (**b**) and TRV infection (**c**,**d**), which are exemplified in this figure by the data obtained in the experiments using fragment 1. (**a**) Virus-induced silencing of the *PARP1* gene in *N. benthamiana* mediated by a PVX vector which contains fragment 1 of the *NbPARP1* gene (PVX-PARP), compared with an empty PVX vector control (PVX-C). Accumulation of PARP1 mRNA was measured using RT-qPCR in inoculated (inoc) and newly emerging systemically infected (sys-N) leaves at 10 dpi. Results from three independent experiments (I, II, III) are shown. (**b**) Effect of PVX-induced PARP1 silencing on the accumulation of PARylated proteins measured by ELISA using a rabbit anti-PAR polyclonal antibody, in the plant leaves shown in (**a**). Results from three independent experiments (I, II, III) are shown. (**c**) Accumulation of TRV RNA1 (measured using RT-qPCR) in inoculated (inoc) and newly emerging systemically infected (sys-N) leaves of PARP-silenced plants at 10 dpi; the same leaves as in (**a**) and (**b**) were analysed. Results from three independent experiments (I, II, III) are shown (**a**–**c**). Data are mean ± SD, *n* = 4 from three independent biological replicates. Analysis of variance and Tukey’s HSD post hoc tests were performed on the data obtained. ** *p* < 0.01; NS, non-significant (**c**,**d**). (**d**) Symptoms induced in plants infected with TRV in PARP1 silenced (PVX-PARP) and non-silenced [PVX-C and (-)PVX] plants.

**Figure 6 viruses-15-01282-f006:**
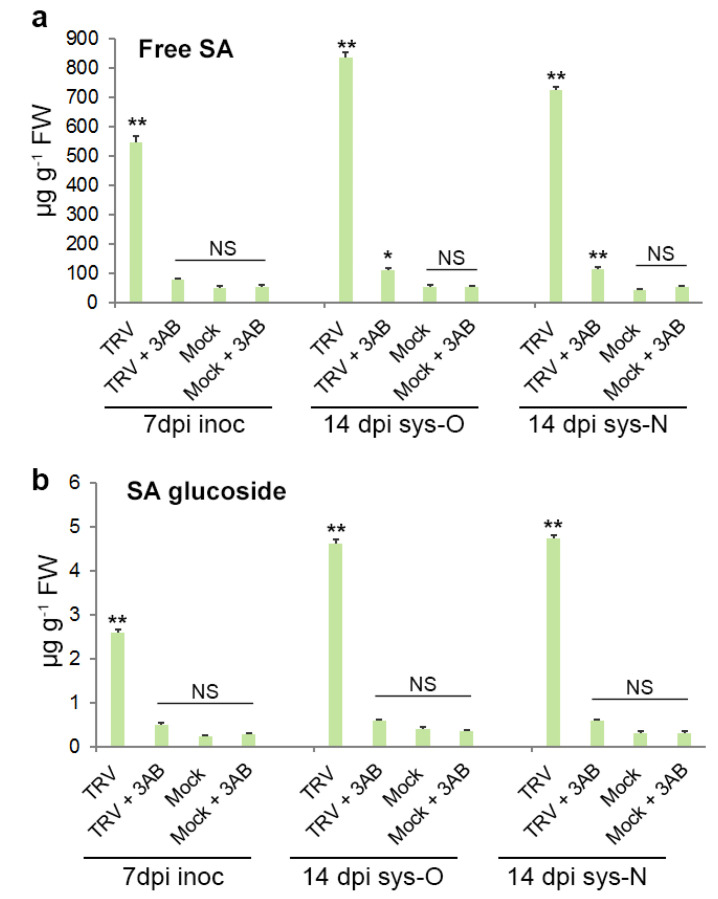
Concentrations of free (**a**) and conjugated (SA-b-glucoside) salicylic acid (SA) (**b**) in inoculated (inoc; 7 dpi), older systemically infected leaves (sys-O; 14 dpi) and newly emerging sys-N; 14 dpi) leaves of *N. benthamiana* plants infected or uninfected (mock-inoculated) with TRV and treated with or without 3AB. Statistical analysis was performed on four independent biological replicates. Data are mean ± SD. Each replicate was composed of samples from three plants pooled together (two leaves per plant). Analysis of variance and Tukey’s HSD post hoc tests were performed on the data obtained. * *p* < 0.05, ** *p* < 0.01. NS, non-significant.

**Figure 7 viruses-15-01282-f007:**
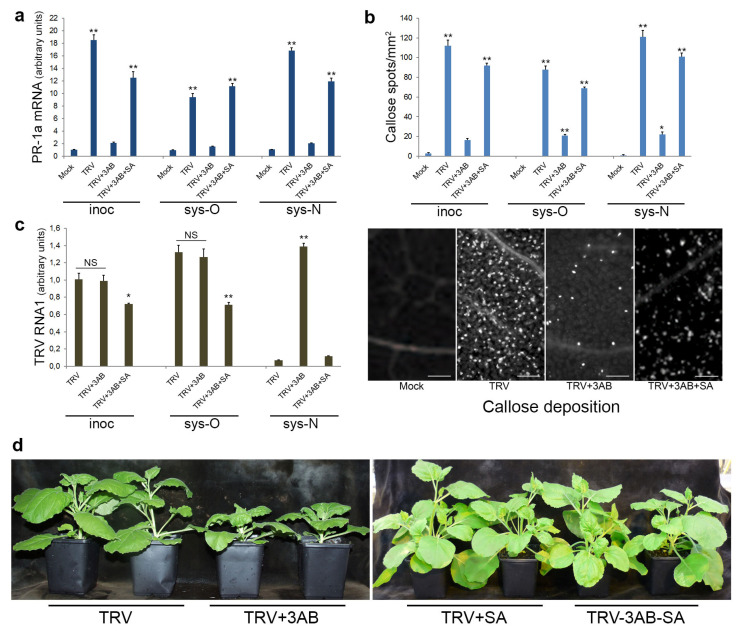
Effect of foliar treatment with salicylic acid (SA) on the transcript level of the PR-1a protein gene (measured using RT-qPCR) (**a**), callose deposition (**b**) and accumulation of TRV (**c**) in inoculated (inoc; 7 dpi), older systemically infected leaves (sys-O; 14 dpi) and newly emerging sys-N; 14 dpi) leaves of *N. benthamiana* plants infected or uninfected (mock-inoculated) with TRV and treated with or without 3AB. Statistical analysis was performed on four independent biological replicates. Data are mean ± SD. Each replicate was composed of samples from three plants pooled together (two leaves per plant). Analysis of variance and Tukey’s HSD post hoc tests were performed on the data obtained. **p* < 0.05, ** *p* < 0.01. (**d**) Symptoms induced in plants infected with TRV after treatment with 3AB and SA as indicated.

**Figure 8 viruses-15-01282-f008:**
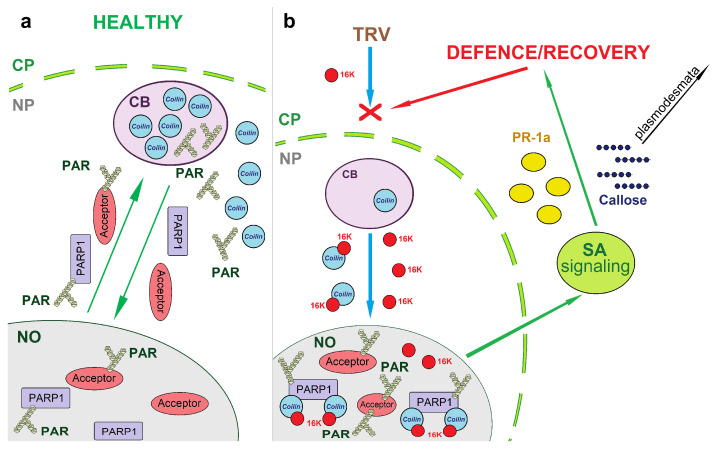
A model of the TRV infection process and the involvements of the 16K protein, coilin and PARP1. In healthy plants (**a**), coilin is located within CBs and the nucleoplasm but is not present in the nucleolus. PARP1, a nuclear protein, modifies the function and subcellular localisation of a variety of nuclear “target” proteins (acceptors) by attaching chains of ADP ribose (PAR) to them. To re-activate these target proteins, PARP1 shuttles them from both the nucleolus (NO) and chromatin (chromatin not shown) to CBs for PAR removal and recycling. Upon TRV infection (**b**), the viral 16K protein is produced in the cytoplasm and is targeted to the nucleus. In the nucleus (CBs and nucleoplasm), the 16K protein interacts with coilin and relocalises it to the nucleolus, which in turn traps PARP1 within this sub-nuclear domain, preventing it trafficking to CBs for PAR cleavage and recycling. This leads to over-accumulation of PAR/PARylated proteins and may enhance accumulation of salicylic acid (SA) and increased elicitation of SA-mediated defence responses (represented here by increased expression of the *PR-1a* gene and by callose deposition). These responses restrict TRV spread into newly emerging leaves, leading to the plant’s recovery from TRV infection. Thus, PARP1 can act as a mediator in a functional link between stress-sensing activities of coilin and SA-mediated antivirus defence.

## Data Availability

The NCBI GenBank accession number for the PARP1 gene reported in this paper is KP771975.

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
