# Peer review of "Plant Poly(ADP-Ribose) Polymerase 1 Is a Potential Mediator of Cross-Talk between the Cajal Body Protein Coilin and Salicylic Acid-Mediated Antiviral Defence"

_viruses, 2023, doi:10.3390/v15061282_

Round 1
Reviewer 1 Report
The MS of Spechenkova describes a role of PARP1 in the crosstalk between CBs and SA-mediated antiviral defense. This is an important finding that deserves publication in Viruses. The MS is clearly written. I have only one minor suggestion: the authors have reported a link between the CB protein coilin and SA signaling. If this can be described in the abstract, the MS may be easier to follow.
Author Response
Dear Reviewer,
Thank you for your posistive repopord.
With regards to your comment:
"...the authors have reported a link between the CB protein coilin and SA signaling. If this can be described in the abstract, the MS may be easier to follow"
we have added a statement showing a link between coilin, PARP and SA signalling (line 26)
Reviewer 2 Report
Dear Viruses Editor,
MS “Plant poly (ADP-ribose) polymerase 1 is a potential mediator of cross-talk 2 between the Cajal body protein coilin and salicylic acid-mediated antiviral defence” presented very interesting results concerning the activation of plant defence responses after TRV infection in WT and coilin silenced N. benthamina plants. Authors showed that after TRV infection, 16K protein (a previously known MP and SSR protein) modulates the interaction between PARP1 and coilin, redistributing both proteins in the nucleous and enhancing SA signaling and PR1 accumulation, what leads to a recovery of virus infection. The data present are robust, and the research design is very good. Data are quite new and very interesting as they are showing a new pathway of SA virus defence induction. I recommend the publication of the present MS after some small corrections.
Authors must review carefully material and methods section. Topic 2.3 at line 160 must describe qRT-PCR methods utilized in the MS however, it contains a copy of the text of the subsequent topic 2.5.
Callose deposition experiments are not explained neither in the material and methods nor in the legend of figure 7.
Some small points are highlighted in the MS pdf.
Best regards

Author Response
Dear Reviewer,
Thank you for your positive report.
We have revised the MS according with your comments:
1.Authors must review carefully material and methods section. Topic 2.3 at line 160 must describe qRT-PCR methods utilized in the MS however, it contains a copy of the text of the subsequent topic 2.5.
Indeed, the section 2.3 was introduced in the text by mistake. We have now replaced this section with the relevant text (lines 183-199)
2. Callose deposition experiments are not explained neither in the material and methods nor in the legend of figure 7.
Callose staining section has now been added to the MS (section 2.12)
3. Some small points are highlighted in the MS pdf.
All these minor points highlighted in the MS pdf have also been addressed properly
Reviewer 3 Report
In this manuscript, the authors elucidated the defense functions of the poly (ADP-ribose) polymerase 1 (PARP1) as a mediator of cross-talk between the Cajal body protein coilin and salicylic acid-mediated to tobacco rattle virus (TRV). The authors determined the physical interactions between CB protein coilin and PARP1 via co-immunoprecipitation, resulting in redistribution of coilin to the nucleolus, finally activating high accumulation of SA and SA-related genes as well as callose deposition, which contributing inhibition of TRV replication. This study implies the mechanism of molecular and biochemical functions of the CB protein coilin as a key candidate protein for plant immunity to TRV, being particularly novel. The work has been performed to an exemplary technical standard, and the data have been presented concisely with commendable lucidity. However, several typographical errors have been found in the manuscript.
There are several major gripes. I have highlighted them in the manuscript.

Author Response
Thank you very much for providing positive report.
We have now addressed all your comments indicated in the MS pdf:
1. Expanations in 2.3 and 2.4 are the same.
Section 2.3. has now been replace by the relevant text
2. Text in sections 2.3, 2.4, 2.5 and Figure legends 2, 4 and 5 are too long and should be shortened.
All these sections and Figure legends have now been shortened.
All other minor syntax errors have been corrected.